# The Role of the NF-kB Pathway in Intracranial Aneurysms

**DOI:** 10.3390/brainsci13121660

**Published:** 2023-11-30

**Authors:** Laurentiu Andrei Blaj, Andrei Ionut Cucu, Bogdan Ionel Tamba, Mihaela Dana Turliuc

**Affiliations:** 1Department of Neurosurgery, “Grigore T. Popa” University of Medicine and Pharmacy, 700115 Iasi, Romania; laurentiu-andrei.blaj@d.umfiasi.ro (L.A.B.); mihaela.turliuc@umfiasi.ro (M.D.T.); 2“Prof. Dr. N. Oblu” Emergency Clinical Hospital, 700309 Iasi, Romania; 3Faculty of Medicine and Biological Sciences, University Stefan cel Mare of Suceava, 720229 Suceava, Romania; 4Advanced Research and Development Center for Experimental Medicine (CEMEX), “Grigore T. Popa” University of Medicine and Pharmacy, 700115 Iasi, Romania; bogdan.tamba@umfiasi.ro; 5Department of Pharmacology, Clinical Pharmacology and Algesiology, “Grigore T. Popa” University of Medicine and Pharmacy, 700115 Iasi, Romania

**Keywords:** intracranial aneurysm, NF-kB pathway, inflammation

## Abstract

The pathophysiology of intracranial aneurysms (IA) has been proven to be closely linked to hemodynamic stress and inflammatory pathways, most notably the NF-kB pathway. Therefore, it is a potential target for therapeutic intervention. In the present review, we investigated alterations in the vascular smooth muscle cells (VSMCs), extracellular matrix, and endothelial cells by the mediators implicated in the NF-kB pathway that lead to the formation, growth, and rupture of IAs. We also present an overview of the NF-kB pathway, focusing on stimuli and transcriptional targets specific to IAs, as well as a summary of the current strategies for inhibiting NF-kB activation in IAs. Our report adds to previously reported data and future research directions for treating IAs using compounds that can suppress inflammation in the vascular wall.

## 1. Introduction

Intracranial aneurysms (IA) are pathologically acquired dilatations of the arterial wall. Their pathophysiology involves several complex mechanisms, including genetic predisposition, hemodynamic stress, inflammation, and vascular remodeling. The prevalence of IA is estimated to be 2–5% in the general population, with an equal distribution among sexes and an annual risk of rupture of 1.4%. Ruptured IA has a 30–50% morbidity and a 50% mortality rate, posing a serious strain to the healthcare system. The pathophysiological mechanisms underlying aneurysm formation, growth, and rupture are still not completely understood, although significant advances have been made in recent years [1]. Aneurysm formation begins with endothelial and hemodynamic dysfunction precipitated by independent risk factors such as smoking, hypertension, and atherosclerosis [2,3]. It is followed by a coordinated inflammatory response to oxidative stress involving macrophages, mast cells, T cells, complement system, and cytokines [4]. These processes will lead to phenotypic modulation of the vascular smooth muscle cells (VSMCs), inducing apoptosis, inflammatory mediators’ upregulation, and the recruitment of other inflammatory cells, resulting in weakening of the arterial wall [5].

The nuclear factor-kappa B (NF-kB) pathway is a key inflammatory signaling pathway that regulates the expressions of various genes involved in cytokine production and cell survival. In VSMCs, NF-kB activation regulates cell survival and apoptosis, contributing to vascular remodeling and the regulation of vascular tone [6]. Also, it regulates the extracellular matrix (ECM) proteins like matrix metalloproteinases (MMPs) and the expressions of inflammatory markers like interleukin 1β and 6 (IL-1β and IL-6) promoting endothelial dysfunction [7,8]. Overall, the NF-kB pathway plays a critical role in the regulation of VSMC function, thus becoming an important pathway in the development and progression of IA by contributing to chronic inflammation and hemodynamic stress.

In this review, we looked at the relationship between the NF-kB pathway and IA in terms of its role in development, progression, and subsequent rupture. This relationship was explored by assessing how the NF-kB pathway impacts endothelial dysfunction, the regulation of VSMCs, and the control of the extracellular matrix, all of which collectively contribute to the formation of IAs.

## 2. Methods

We conducted a literature search of the MEDLINE/PubMed and EMBASE databases to identify studies related to the role of the NF-kB pathway in intracranial aneurysms. The search was performed on September 2023 with the keywords “intracranial aneurysm”, “NF-kB pathway”, “vascular smooth muscle cell”, “extracellular matrix”, and “endothelial dysfunction”. Two investigators (LAB and AIC) independently screened titles and abstracts. Discrepancies in the study selection were discussed with all authors and a consensus was reached. 

The eligibility criteria consisted of: (1) studies reporting the relationship between the NF-kB pathway and intracranial aneurysms; (2) studies reporting the role of the NF-kB pathway in vascular smooth muscle cells, the extracellular matrix, and endothelial dysfunction; (3) studies in the English language; and (4) original research articles and reviews (editorials and letters were excluded). 

## 3. Results

We screened 654 articles published between 1980 and September 2023. After the first screening by abstract, we excluded 12 duplicate studies, 224 unrelated to the topic, and 148 not relevant. A total of 193 articles were excluded after the full-text screening identified as irrelevant. We included 77 articles based on the eligibility criteria.

Most authors have employed similar models for IA induction, which was first described by Hashimoto et al. in rats [9]. The basic concept involves inducing hemodynamic stress to the arterial walls through unilateral common carotid ligation, unilateral renal artery ligation, and a high-salt diet. The results consisted of IA induction at the anterior cerebral artery/olfactory artery bifurcation. To this end, we can conclude that the results are specific to anterior circulation aneurysms; however, no differences were found between potential patient populations or subtypes of IA in the available research.

Regarding the timing of the intervention, most studies chose to start treatment immediately after aneurysm induction, the day after, or one month after. No explanation was provided regarding the optimal stage for implementing therapeutic strategies.

## 4. Discussion

### 4.1. Overview of the NF-kB Pathway

The NF-kB pathway represents a family of regulated transcription factors that are activated concurrently, leading to gene transcription [10]. It was first discovered in 1986 by Baltimore et al. [11]. The transcriptional factors involved are homodimers or heterodimers of Rel proteins like p65/p50, p52, and its precursor p100 [12]. The intermediate proteins involved in the NF-kB activation pathway are represented by tumor necrosis factor receptor type 1-associated death domain (TRADD), tumor necrosis factor receptor-associated factor 2 (TRAF2), NF-kB-inducing kinase (NIK), mitogen-activated protein kinase/ERK kinase-1 (MEKK), and IκB kinase (IKK) [13]. Since its discovery, the NF-kB pathway has been shown to play a critical role in regulating the expressions of genes involved in immune function, inflammation, and cellular growth control [14].

The NF-kb pathway can be activated by various stimuli, including tumor necrosis factor α (TNF-α), interleukin 1 (IL-1), oxidative stress, lipoproteins from Gram-negative bacteria, activated B-cells, viral RNA, and hemodynamic pressure [15,16,17]. Given its responsiveness to a wide range of stimuli, it is a significant and intricate process. This complexity has led to its division into canonical and non-canonical pathways.

The canonical NF-κB pathway (Figure 1) is well characterized and is mediated by the IKK complex in response to pro-inflammatory signals [18]. The IKK complex consists of two catalytic subunits, IKKα and IKKβ, and a regulatory subunit, NEMO (also known as IKKγ) [19]. IKKα is structurally related to IKKβ and both contain a serine/threonine kinase domain at their N-terminus and protein–protein interaction motifs, including a leucine zipper (LZ) and a helix–loop–helix (HLH) at their C-terminal portion. IKKγ lacks catalytic activity and contains several coiled-coil motifs that mediate oligomerization [20]. In the absence of an external stimulus, NF-kB dimers remain in an inactivated state in the cytoplasm. IKKα/IKKβ dimers combine with IKKγ to activate the IKK complex. Once activated, IKKβ phosphorylates IkBα and associates with βTrCP. This initiates ubiquitination of IkBα and its subsequent degradation by the cytosolic proteasome. The inactivated form of the NF-kB dimer is composed of p50 and p65 proteins, coupled with IκBα. Once the latter is removed, the p50/p65 dimer is free to enter the nucleus and bind to its target genes [20]. The p50 subunit is known to interact specifically with Bcl-3 protein and influences the apoptotic behavior of the cell [21]. The p65 subunit plays a crucial role in inflammation and is known to interact with protein kinase C-δ (PKCδ) in vascular smooth muscle cells, regulating proinflammatory chemokine expression and modulating inflammation [22]. In addition, it binds to the Ccl2 and Ptgs2 genes which encode MCP-1 and COX-2, respectively [23].

The NF-kB non-canonical pathway (Figure 2) is mediated by NF-kB-inducing kinase (NIK) and is activated when ligands bind to TNFR, CD40, BAFF, and TLβR receptors [24]. NIK was first identified as a mitogen-activated kinase (MAP3K) and is the first component of the non-canonical pathway [25]. It remains in an inactivated form when bonded to the TRAF3/TRAF2/cIAP complex. Once the ligand–receptor complex is formed, the TRAF3/TRAF2/cIAP complex is recruited and NIK is activated. NIK directly phosphorylates IKKα but does not influence IKKβ [24]. IKKα induces the proteolysis of p100 to p52, generating p52/RelB dimers in order to activate NF-kB regulated anti-apoptotic bcl-2 and bcl-xl genes [26]. In the end, the transcription of Bcl-2 and Bcl-xl genes will lead to an increased cell survival [27].

### 4.2. Stimuli That Activate the NF-kB Pathway in IA Pathology

The NF-kB pathway can be activated in endothelial cells by a variety of stimuli, which are presumed to be causative factors in intracranial aneurysm pathology. Some of these stimuli include inflammatory cytokines (TNF-α and IL-1β), Toll-like receptor agonists, and hemodynamic stress. All of them have a role in vascular smooth muscle motility, extracellular matrix remodeling, and endothelial dysfunction.

In an experimental study, Aoki et al. revealed that the increased TNF-α content in IA lesions is caused by an increase in hemodynamic stress. They also proved that the incidence of IA was significantly suppressed in TNFR1-heterozygous and TNFR1-deficinet mice compared to that in wild-type mice, thus clarifying the crucial role of TNF-alpha-TNFR1 signaling in IA formation [28].

Procollagen types I and III expressions are decreased in cerebral aneurysm walls, possibly due to the activation of the p65 subunit of the NF-kB pathway by IL-1β. This, in turn, leads to the downregulation of procollagen types I and III and lysyl oxidase (LOX). This process can persist for up to three months following aneurysm induction, resulting in a loss of structural support [29]. A loss of collagen fibers and increased elastic capacity in the aneurysm wall leads to the formation of a ruptured zone, thus initiating the development and rupture of IA [30]. 

Toll-like receptors (TLR) are involved in another pathway for NF-kB activation. The expression of TLR4 was present in the endothelial cells and adventitia in human unruptured cerebral aneurysms, whereas rats presented it only in endothelial cells. The experimental data show that TLR4 expression reached its peak 1 month after aneurysm induction and decreased to the level before aneurysm induction 3 months after [31]. The expressions of TLR2, MyD88, and NF-kB p65 in patients with intracranial aneurysms are significantly higher compared to normal controls, which is why Zhang et al. proposed the TLR2/4-MyD88-NF-kB signaling pathway to be involved in the pathogenesis of intracranial aneurysms [32].

The link between hemodynamic stress, cerebral aneurysm formation, and the NF-kB pathway is the PGE2-EP2 signaling function. Shear-stress-induced PGE2-EP2 receptor signaling activates NF-kB in endothelial cells at an early stage and contributes to chronic inflammation in the arterial walls for cerebral aneurysm formation [33]. Another mechanism is based on variable mechanosensory activation. In human intracranial aneurysms, low shear stress clearly identified regions with high NF-kB activity, whereas disturbed flow was associated with randomly aligned cells and high NF-kB activation [34].

### 4.3. Transcriptional Targets of NF-kB Pathway in IA Pathology

The NF-kB pathway controls the expression of a variety of genes involved in the immune response, cell survival, proliferation, and inflammation. Transcriptional targets (Figure 3) depend on the signaling molecules involved and are tightly regulated. Cytokine release is a major consequence of this pathway. Activated NF-kB regulates the expression levels of IL-1β, IL-6, and TNF-α via the canonical pathway and phosphorylation of IκB. Mezzasoma et al. found that the NF-kB/ERK1/2 pathway is involved in the processing and release of IL-1β [35]. This creates a positive feedback loop that promotes chronic inflammation [36,37]. 

The NF-kB pathway also regulates the expression of adhesion molecules. NF-κB activation leads to the expression of adhesion molecules such as E-selectin, vascular cell adhesion molecule-1 (VCAM-1), and intracellular adhesion molecule-1 (ICAM-1) [38]. This close relationship is due to the fact that the promoter regions of ICAM-1, VCAM-1, and ELAM-1 contain NF-kB binding sites. The treatment of endothelial cells with TNF increases the adhesion of monocytes to the endothelial surface, thus amplifying the inflammatory response [39].

Regarding chemokines, the NF-kB pathway plays a crucial role in the recruitment and shaping of the cellular microenvironment. Also, this could explain the active recruitment of inflammatory cells in the aneurysm wall. Monocyte chemoattractant protein 1 (MCP-1) and macrophage inflammatory protein 1α (MIP-1α) can be regulated by protein kinase C-δ (PKCδ) through cytosolic interactions with the NF-kB subunit p65 in vascular smooth muscle [22]. In addition, MCP-1 expression can be influenced by reducing the upstream phosphorylation of IκB, as well as the activation of the NF-kB promoter [40]. Chalouhi et al. found high plasma concentrations of chemokines and chemoattractant cytokines in the lumen of 18 human IA, whereas Aoki et al. found that MCP-1 expression was upregulated in the aneurysm walls in the early stages of aneurysm formation [41,42].

One of the major roles of the NF-kB pathway is the regulation of anti-apoptotic proteins Bcl-2, Bcl-xL, and apoptotic protein A20. Bcl-2 is regulated by p53 protein binding to its p1 promoter [27,43]. A20 regulates the immune response and controls the apoptotic pathway induced by TNF-α. Its function is mediated via the NF-kB dependent alteration of cIAP1/2 in the non-canonical signaling [44]. In IA, the upregulation of the Bcl-2 gene by MiR-140 knock-out alleviates its development and progression [45]. Therefore, a regulated Bcl-2 by the NF-kB pathway could represent a potential therapeutic target for IA.

### 4.4. Role of NF-kB Pathway in Vascular Smooth Muscle Cell

The NF-kB pathway plays an important role in the vascular smooth muscle cell (VSMC) by being involved in the pathogenesis of vascular diseases, such as atherosclerosis, restenosis, and hypertension. It regulates VSMC by inducing proliferation, apoptosis, and differentiation (Table 1). A vascular smooth muscle cell is a highly specialized cell that has a crucial role in regulating vascular tone and blood pressure. Its primary function is contraction and relaxation, which is regulated by contractile proteins, ion channels, and signaling molecules [6]. IA growth is marked by the ongoing structural weakness of nearby smooth muscle cells, although the exact process remains uncertain. Liu et al. demonstrated that consistent cyclic mechanical stretching prompts alterations in the aneurysm tissue by exhibiting fewer VSMCs and decreased collagen type IV and VI. Preserving the well-being of VSMC offers an additional avenue for therapeutic intervention aimed at preventing the progression of IA [46].

In VSMCs, both the canonical (p65) and non-canonical (p52) NF-kB signaling pathways can be activated. Exposure to the receptor activator of nuclear factor kB-ligand (RANKL) can increase both NF-kB/p52 and NF-kB/phosphor-p65 levels, whereas treatment with tumor necrosis factor-related apoptosis-inducing ligand (TRAIL) has been proven to significantly increase NF-kB/Phospho-p65 levels [47]. Another way to activate the NF-kB pathway in vascular smooth muscle cells is through *Chlamydophila pneumoniae*-induced ICAM-1 expression. Vielma et al. proved that, in human aortic endothelial cells, NF-kB activation is protein kinase C (PKC)-dependent when subjected to *C. pneumoniae*-infected endothelial cells [48]. An activated NF-kB pathway leads to a pro-inflammatory status in the vascular wall. In a study on cell cultures, Currie et al. demonstrated that heat shock factor-1 (HSF-1) exacerbates Ang-II-induced inflammation via the NF-kB pathway. A countermeasure to it is heat shock protein 27 (Hsp27) that regulates the phosphorylation of p65 subunit of NF-kB in the Ang II-induced signaling pathway of NF-kB by suppressing the IKK complex [49,50].

The calcification of VSMCs, a byproduct of chronic inflammation, leads to a loss of elasticity and increased stiffness of the vessel wall. Excessive glucose concentrations can activate NF-kB and trigger pro-calcific effects in VSMCs, which may actively augment vascular calcification [51]. One way to prevent such a development is zinc supplementation. In a study by Voelkl et al., zinc supplementation upregulated the endogenous inhibitor of NF-kB, zinc-finger protein TNFAIP3, and decreased IkBα phosphorylation, upregulated IkBα protein abundance, and triggered the p65 subunit with decreased NF-κB-dependent transcriptional activity [52].

The survival of VSMCs depends on the NF-κB/Rel transcription factors. They regulate the expression of Bcl-xL and Bfl-1/A1, which protect the cells against death. Romano et al. proved that TNF-α enhanced the basal expression of Bcl-xL twofold via the NF-kB pathway and it induced the expression of Bfl-1/A1 [53]. TNF-α-induced NF-kB pathway activation can be managed by A20 protein (also known as tumor necrosis factor α-induced protein 3–TNFAIP3). Utilizing its deubiquitinating activity, A20 removes Lys63 polyubiquitin chains from the NF-kB essential modulator (NEMO), TNF receptor-associated factor 6 (TRAF6), and receptor-interacting protein ½ (RIP1/2) to suppress NF-kB activation in vascular smooth muscle cells [54].

**Table 1 brainsci-13-01660-t001:** Overview of NF-kB pathway in vascular smooth muscle cells.

Trigger	Mechanism	Effect	Treatment	References
RANKL	p52 pathway	Pro-calcific effect	TRAIL	[47]
*C. pneumoniae*	PKC dependent pathway	Increased expression of ICAM-1	Not specified	[48]
HSF-1	Ang-II-induced inflammation	Activation of proinflammatory transcription factors	Hsp27	[49,50]
Hyperglycemia	Activation of SGK 1	Vascular calcification	Zinc supplementation	[51,52]
TNF-α	p65 pathway	Increased expression of Bfl-1/A1 and Bcl-xL	A20 protein	[54]

RANKL, receptor activator of nuclear factor kB-ligand; TRAIL, tumor necrosis factor-related apoptosis-inducing ligand; *C. pneumoniae*, *Chlamydophila pneumoniae*; PKC, protein kinase C; ICAM-1, intracellular adhesion molecule-1; HSF-1, heat shock factor-1; Ang-II, angiotensin II; Hsp27, heat shock protein 27; SGK 1, glucocorticoid-inducible kinase 1; and TNF-α, tumor necrosis factor α.

### 4.5. Role of NF-kB Pathway in Extracellular Matrix Remodeling

Extracellular matrix (ECM) remodeling is a complex process involving the synthesis, deposition, and degradation of ECM proteins. The ECM provides structural support and plays a critical role in cell signaling, migration, proliferation, and differentiation. ECMs’ enzymes are a group of proteolytic enzymes that are involved in the degradation and remodeling of EMCs. Examples of such enzymes are matrix metalloproteinases (MMPs), serine proteases, and cysteine proteases. MMPs are regulated by various signaling pathways, including the NF-kB pathway. 

The NF-κB-binding site is present in the promoter of the MMP-9 gene and an NF-κB-like element in the promoter of the MMP-1 gene. Using an over-expression of IkBα, Bond et al. proved that the NF-kB pathway is required for the cytokine upregulation of MMP-1, -3, and -9 in VSMCs [7,55]. In cardiac myoblast H9c2 cells, TNF-α increased the mRNA expression and protein activity of MMP-9 [56]. Also, MMP-2, MMP-9, VEGF, ROS, IL-1β, and IL-6 were significantly increased in TNF-α-induced endothelial cells via the NF-kB pathway [8]. In a study on fibroblasts, Xu et al. demonstrated that MMP-1 expression was decreased when the NF-kB pathway was inhibited by the peptide inhibitor SN50 [57]. TNF-α exposed cells showed increased NF-κB activity and increased ratios of MMP-2/TIMP-2, and MMP-9/TIMP-1 in airway smooth muscle cells (ASM). In turn, extracellular matrix deposition can be reduced by suppressing the NF-kB pathway via the estrogen receptor β (ER-β) [58]. In IA, MMP-2 and -9 are two important factors that promote progression; therefore, they can become candidates for treatment targets via the NF-kB pathway [59]. 

A connection between TIMP-2 and the NF-kB signaling pathway has been established in melanoma cell lines and lung epithelial cells. Sun et al. proved that tissue inhibitor of metalloproteinase-2 (TIMP-2) overexpression increased NF-kB transcriptional activity, via decreasing the inhibitory molecule IkBα [60]. A more in-depth analysis was conducted by Lizarraga et al. In their study, they concluded that NF-kB activity was increased by exposure to TIMP-2, with maximal activity being achieved after 24 h and rapidly decreasing by 48 h. Also, IkB1α was downregulated 24 h after exposure to TIMP-2, while IkBβ levels were only slightly decreased. Bcl-3 expression diminished 24 h after TIMP-2 exposure and increased after 48 h [61]. It can be inferred from these findings that the primary factor is the inhibition of IkB1α.

Regarding MMP-3, Sanchavanakit et al. observed that the pretreatment of murine cementoblast cell lines with pyrrolidine dithiocarbamate (PDTC), an inhibitor of NF-kB or p38 MAPK, resulted in MMP-3 suppression [62]. Also, in a study on glioma cells, the upregulation of the Bmi-1 gene promoted cell migration and invasion via the NF-kB-mediated upregulation of MMP-3. Another NF-kB inhibition drug, Resveratrol, significantly reduced the level of MMP-3 [63,64]. 

### 4.6. The NF-kB Pathway in Endothelial Dysfunction

Endothelial dysfunction is a condition in which the endothelium loses its normal function and is unable to regulate vascular tone and permeability. Under normal conditions, the NF-kB pathway is tightly regulated in endothelial cells. However, under various conditions, endothelial dysfunction takes over and can lead to an uneven expression of adhesion proteins, loss of permeability, and vasodilatory impairment. Also, damage to the endothelial cell layer is cited as the first event in IA formation [65].

The activation of the NF-kB pathway in endothelial cells has been shown to upregulate hypoxia inducible factor-1 (HIF-1) via a positive feedback loop. This, in turn, causes endothelial cell dysfunction, cell proliferation, angiogenesis, and inflammation [66]. In addition, dermal microvascular endothelial cell (HMECs-1) permeability is influenced by the hemoglobin (Hb)-induced NF-kB pathway, which is under the control of the TLR signaling pathway [67].

In a randomized, double-blind, placebo-controlled crossover study of 14 non-diabetic overweight or obese adults, Pierce et al. demonstrated that activating the NF-kB pathway via the stimulation of oxidative stress led to marked impairments in baseline vascular function by influencing endothelium-dependent dilatation [68]. Vascular tone can be influenced by the NF-kB-mediated COX-2/iNOS pathway. Using an androgen receptor agonist, Gonzales et al. found that dihydrotestosterone enhanced NF-kB activation and increased cyclooxygenase-2 protein levels in rats. These data suggest an increase in the basal production of vasodilatory factors [69]. Vasodilatation will, in turn, create a low endothelial wall shear stress that may facilitate the growing phase and initiate the rupture of IA [70].

The NF-kB pathway can modulate endothelial cell death and increase cerebral vessel permeability. The selective deletion of the NF-kB essential modulator (NEMO) or the upstream kinase Tak1 in brain endothelial cells resulted in cerebral hypoperfusion and disruption of the blood–brain barrier (BBB). This proved that TAK1-NEMO signaling has a protective role in cerebral vessels via the NF-kB pathway [71]. Another pathway to disrupt the BBB is via the NOTCH3/NF-kB signaling pathway. IL-1β induced the nuclear translocation of NF-kB/p65, upregulated MMP-9 expression in pericytes, and increased permeability in the BBB [72].

Regarding the influence of the NF-kB pathway on adhesion molecules, the inhibition of NF-kB pathway using amino acids like glycine, histidine, and cysteine has been proven to reduce E-selectin expression and IL-6 production in human coronary arterial endothelial cells (HCAECs) stimulated with TNF-α [73]. Also, increased lipid peroxides in preeclamptic plasma stimulate ICAM-1 expression through a NF-kB-mediated mechanism. This hypothesis is supported by the findings that vitamin E and N-acetyl cysteine not only inhibit endothelial cell NF-kB activation by preeclamptic plasma, but also ICAM-1 expression, leading to endothelial dysfunction [74]. A similar relationship between ICAM-1 and NF-kB activation has been found in cells subjected to angiogenic basic fibroblast growth factor (bFGF/FGF-2). bFGF inhibits the TNF-mediated activation of NF-kB by blocking the phosphorylation and degradation of IkBα and decreases ICAM-1 expression. The end result is a decrease in cell adhesion and increased permeability [75]. The latter has also been observed in the wall of the IA compared with paired healthy cerebral arteries [76].

TNF-α-activated endothelial cells stimulate VSMCs and immune cell proliferation and promote the early and late apoptosis of VSMCs. This leads to an increase in inflammation in the vascular wall and promotes aneurysm formation [77].

### 4.7. Current Strategies for Inhibiting NF-kB Activation

The management of IA remains challenging due to the limited understanding of its pathogenesis. The current working hypothesis has been reduced to an inflammatory injury coupled with hemodynamic stress, placing the NF-kB pathway at the forefront of the discussion. As such, the inhibition of the NF-kB pathway has emerged as a promising strategy for the treatment and prevention of IA (Table 2). Attempts have been made with both pharmaceutical and natural extracts, but so far, the results have only been in the experimental phase.

Research results suggest that statins can inhibit the NF-kB pathway. The anti-inflammatory action of statins derives from suppressing TLR4/MyD88/NF-kB signaling by decreasing the expression of TLRs 2 and 4 [78]. Pitavastatin suppresses IA progression by reducing the number of DNA-binding forms of the NF-kB p65 subunit. Moreover, pitavastatin treatment can cause the regression of degenerative changes in preexisting IA walls [79]. Also, treatment with simvastin suppresses the development of IAs and has a preventive effect on the progression of preexisting IAs [80]. 

Another commonly used drug is aspirin. It has been proven that aspirin significantly reduced the degeneration of aneurysm walls by inhibiting macrophages-mediated chronic inflammation and dramatically inhibiting the expression of NF-kB and monocyte chemoattractant protein-1 (MCP-1) in the aneurysm wall [81]. However, treatment with aspirin did not appear to alter the histopathological findings after aneurysm formation, suggesting that it would have its greatest effect during the acute inflammatory phase [82].

Nifedipine is a widely used calcium antagonist and it has been proven to inhibit TNF-α induced reactive oxygen species (ROS) generation, as well as decrease the NF-kB activity in cultured fibroblasts. In rats, nifedipine inhibited DNA binding NF-kB in aneurysmal walls, preventing the enlargement and degeneration of the aneurysmal walls of preexisting IAs [83].

Regarding the prevention of IA formation, zinc supplementation significantly increased the expression of the anti-inflammatory signaling protein A20, an inhibitor of the nuclear factor kB (NF-kB) pathway, in rat IAs. Zinc administration may prevent the growth of rat IAs by inducing the A20-attributed inactivation of NF-kB signaling [84]. However, long-term treatment with zinc (more than 3 months) can develop resistance and enhance the activity of the NF-kB pathway, which can lead to the activation of metastatic mechanisms [85].

The TNF-α inhibitor, Etanercept, significantly reduced aneurysm formation in rats. It suppressed the expressions of NF-kB, iNOS, MMP-9 mRNA, and phospho-IKKα and phospho-IKKβ mRNAs. There was no significant difference at 1 month or 3 months after aneurysm induction between the control group, low-dose etanercept group, and high-dose etanercept group [86]. Side effects from using TNF-α inhibitors have been observed in individuals with rheumatoid arthritis and include potential cancer risks, however, the study results remain controversial [87]. Also, the STAT3 inhibitor, BP-1-102, can reduce the expression of inflammatory factors and MMPs bound to NF-kB by inhibiting the activation of the JAK/STAT3/NF-kB pathway, and then restore the vascular wall elastin to reduce blood pressure, thereby treating aneurysms in mice [88]. Another compound that has proven its efficacy, although still experimental, is represented by bone marrow mesenchymal stem cells (BMSCs). Human BMSC-derived exosomes inhibited the activation of the phosphatidylinositol-3 kinase (PI3K)/protein kinase B (Akt)/ nuclear factor-kappa B (NF-kB) signaling pathway and maintained Th17/Treg balance, which, in turn, suppress aneurysm formation [89].

Natural extracts have been shown to possess anti-inflammatory properties. Resveratrol, a polyphenol compound widely found in plants, such as *Polygonum cuspidatum*, *Cassia tora*, grapes, and peanuts, can inhibit IA formation in mice subjected to induced hypertension by downregulating the NF-kB pathway [90]. Also, Tanashione IIA (Tan IIA) is an anti-inflammatory component isolated from a traditional Chinese medicine called Danshen (*Salvia miltiorrhiza Bunge*). Tan IIA treatment prevented the process of macrophage infiltration and the degeneration of aneurysmal walls due to the inhibition of NF-kB and MCP-1 expression in rats [91].

**Table 2 brainsci-13-01660-t002:** Therapeutic targets for NF-kB pathway inhibition.

Compound	Animal Model/Cell Culture	Mechanism of Action	Effect	Year	References
Pitvastin	Rats	Reduced binding to DNA for p56 subunit	Smaller aneurysm size, lower expression of MCP-1, VCAM-1, MMP-9, IL-1β, and iNOS	2009	[79]
Simvastin	Rats	Not specified	Reduced expression of MCP-1, VCAM-1, MMP-2, and MMP-9	2008	[80]
Aspirin	Rats	Not specified	Reduced expression of MCP-1, VCAM-1	2015	[81]
Nifedipine	Rats	Inhibited translocation of p65 subunit into the nucleus	Reduced expression of MCP-1 and MMP-2	2008	[83]
Zinc	Rats	Increased production of A20 protein	Smaller aneurysm size	2021	[84]
Etanecerpt	Rats	TNF-α inhibitor	Smaller aneurysm size and thicker internal elastic membrane	2014	[86]
BP-1-102	Mice	STAT3 inhibitor	Reduced expression of TNF-α, IL-1β, IL-6, MCP-1, IL-10, and MMP	2020	[88]
BMSCs	Cell culture/rabbits	Inhibition of KLF5 and suppression of PI3k/Akt/NF-kB pathway	Increased eNOS mRNA expression, decreased iNOS mRNA, MMP-2, and MMP-9 expression	2020	[89]
Resveratrol	Cell culture	MMP-3 inhibitor	Inhibits cell apoptosis by increasing the expression of Bcl-2 protein, decreased iNOS and NO expression	2014	[64]
	Mice	Not specified	Lower IA incidence, reduced expression of MMP-2, and MMP-9	2022	[90]
Tanshione IIA	Rats	TNF-α inhibitor	Reduced expression of MCP-1, MMP-2 and MMP-9	2019	[91]

STAT3, signal transducer and activator of transcription 3; KLF5, kruppel-like factor 5; PI3k, phosphoinositide-3-kinase; Akt, protein kinase B; NF-kB, nuclear factor kB; MCP-1, monocyte chemoattractant protein-1; VCAM-1, vascular cell adhesion molecule 1; MMP, matrix metalloproteinase; TNF-α, tumor necrosis factor α; IL-1β, interleukin 1β; IL-10, interleukin 10; iNOS, inducible nitric oxide synthase; eNOS, endothelial nitric oxide synthase; and NO, nitric oxide.

To our knowledge, there are no studies in which a combination of therapeutic strategies has been employed, although Aoki et al. proved that the synergic activation of the NF-kB pathway with prostaglandin E_2_ and TNF-α increases the expression of inflammatory markers [23]. We believe that the next step in the inhibition of the NF-kB pathway is a combination of current strategies to prevent the progression and rupture of IA. The inhibition of the NF-kB pathway has proven advantageous in diverse neurovascular disorders, such as intracranial stenosis, moyamoya disease, and ischemic stroke. Consequently, the therapeutic approaches outlined above hold potential for applications in managing these specific conditions [92]. 

### 4.8. Clinical Trials Targeting Inflammation for Treating IA

As of the writing of this review, we identified several clinical trials targeting the reduction of inflammation to treat IA. Of great interest is the ongoing Statin Treatment for Unruptured Intracranial Aneurysms Study (STUDIES), where researchers are looking to identify whether oral atorvastatin administered daily for six months will reduce the inflammatory signs in the wall of saccular aneurysms on high-resolution magnetic resonance imaging (HRMRI) [93]. The same protocol was used in the ATREAT-VBD trial that focused on unruptured intracranial vertebrobasilar dissecting aneurysms that did not benefit from surgery [94]. Aspirin has been used in two clinical trials so far. Hassan et al. conducted a prospective study of 11 patients using low-dose aspirin (81 mg daily for 3 months) and showed promising results [95]. A high dose of aspirin (300 mg daily for 3 months) is currently ongoing in the Unruptured Intracranial Aneurysm Aspirin Trial (UIAAT) [96]. Regarding ruptured IA, the ongoing FINISHER trial is investigating the use of dexamethasone in aneurysmal subarachnoid hemorrhage to reduce delayed cerebral ischemia [97].

### 4.9. Future Research Directions for Treating IA

Although recent studies have investigated the potential of targeting the NF-kB pathway for IA treatment, there is still much to be learned regarding the specific mechanisms involved. Potential avenues for future investigation already discussed in this review may consist of the peptide inhibitor SN50, pyrrolidine dithiocarbamate (PDTC), vitamin E, and N-acetyl cysteine. Further research may focus on anti-cold induced RNA binding protein (CIRP) antibody to suppress vascular wall inflammation via the TLR4/NF-kB pathway. Promising results have been obtained for abdominal aortic aneurysms in rats [98]. Also, microRNA(miR)-195 suppressed abdominal aortic aneurysms through the TNF-α/NF-kB and VEGF/PI3K/Akt pathway [99]. Prostaglandin E receptor subtype 2 (EP2) antagonists can be another potential therapeutic target for aneurysm suppression by inhibiting the prostaglandin E2-EP2-NF-kB signaling in macrophages [23].

Traditional Chinese medicine has offered anti-inflammatory extracts that are rumored to inhibit the NF-kB pathway. Paeonol, a major phenolic ingredient extracted from peony bark, has been proven to alleviate inflammatory disorders and aneurysm progression, while Daidzein attenuates abdominal aortic aneurysm in angiotensin II-induced mice via NF-kB, p38MAPK, and TFG-β1 [100,101].

Future research should focus on whether the inhibition of the NF-kB pathway might be beneficial for preventing IA recurrence after endovascular coiling or surgical clipping. To this end, imaging modalities such as the auto-segmentation of tomography images or hybrid machine learning systems may represent future directions for the better selection of patients that benefit from targeting the NF-kB pathway [102,103].

### 4.10. Limitations

Current therapeutic strategies have only been employed in preclinical settings, with the exception of statins and aspirin. Potential challenges could arise from translating them to clinical treatments, mainly because there is little variability in the preclinical models used, the sample sizes are small, and the research is conducted under standard conditions, unlike in a clinical scenario. Regarding efficacy and safety, aspirin was the only drug with preliminary results in patients presenting with IA. In addition, no known interactions or contraindications between NF-kB-pathway-targeted therapies and other medications commonly used in IA treatment have been reported. Another limitation is the limited number of clinical trials investigating the effect of NF-kB pathway inhibition; therefore, to our knowledge, no long-term outcomes have been reported in patients with IA.

## 5. Conclusions

The NF-kB signaling pathway is an important mediator of the pathophysiology of intracranial aneurysms through its role in vascular smooth muscle cell proliferation, extracellular matrix remodeling, and endothelial dysfunction. The identification of various stimuli and transcriptional targets has led to the development of various strategies to inhibit their activation. Most notably, statins and aspirin are currently used in human trials and hold great promise. However, there is still much to be learned about the precise role of NF-kB in IA pathology. Overall, the NF-kB pathway is a promising therapeutic target for aneurysm formation and prevention and is a reliable source for identifying novel therapeutic targets.

## Figures and Tables

**Figure 1 brainsci-13-01660-f001:**
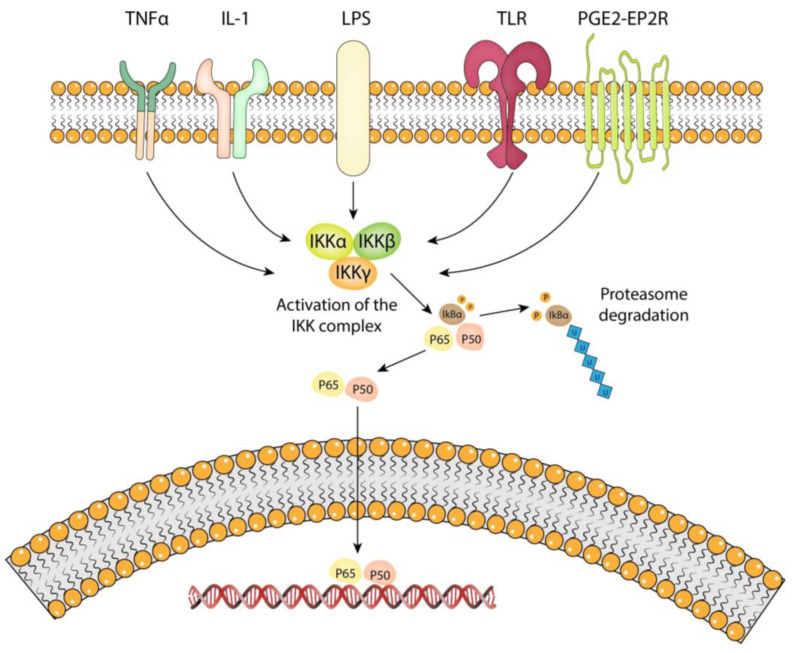
Canonical NF-kB pathway. Activation of the NF-kB pathway through the TNF-α, IL-1, LPS, TLR, and PGE2-EP2 receptors induces the activation of the IKK complex, which is composed of two catalytic subunits, IKKα and IKKβ, and a regulatory subunit IKKγ. Once activated, IKKβ phosphorylates IkBα and initiates its subsequent degradation, leaving the p50/p65 dimer free for entry into the nucleus. The illustration is created by the authors using Adobe Illustrator v2020.

**Figure 2 brainsci-13-01660-f002:**
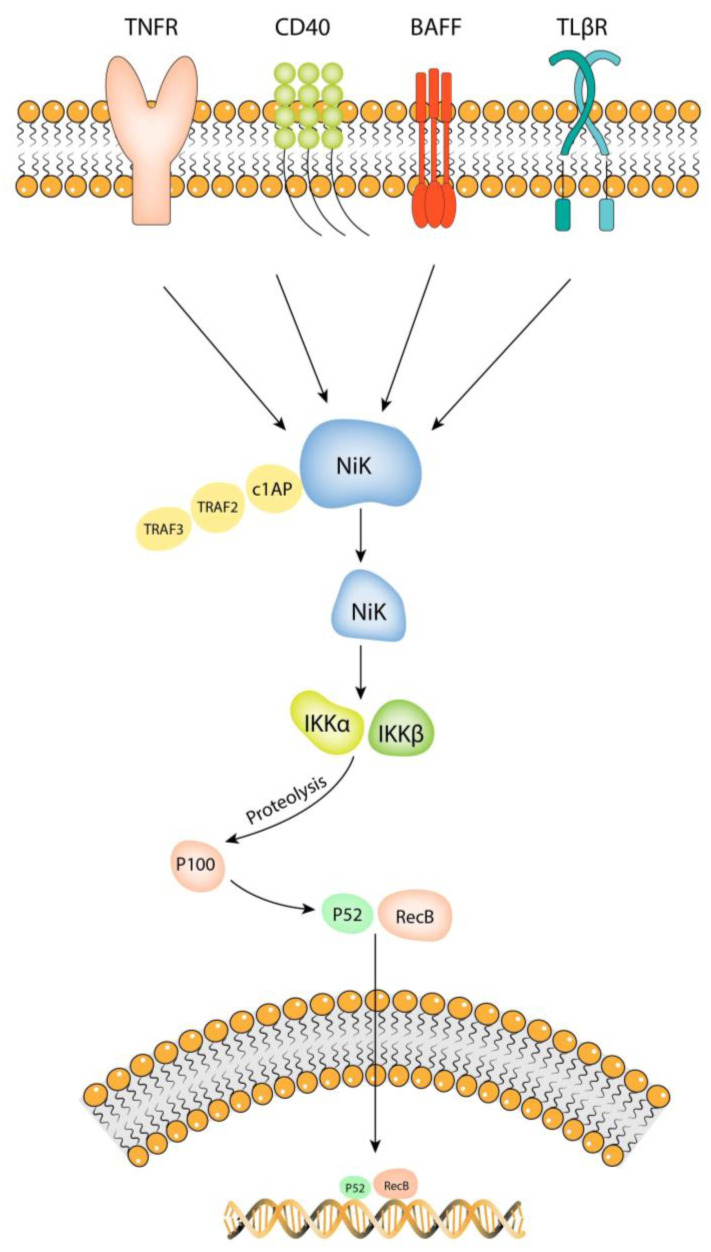
Non-canonical NF-kB pathway. Activation of the non-canonical pathway via the ligand binding to TNFR, CD40, BAFF, and TLβR receptors will lead to recruitment of the TRAF3/TRAF2/cIAP complex, leaving the NIK unrestricted. Once activated, NIK will directly phosphorylate IKKα. This causes IKKα to induce the proteolysis of p100 to p52, generating p52/RelB dimers. The P52/RelB dimer enters the nucleus and regulates the bcl-2 and bcl-xl genes. The illustration is created by the authors using Adobe Illustrator v2020.

**Figure 3 brainsci-13-01660-f003:**
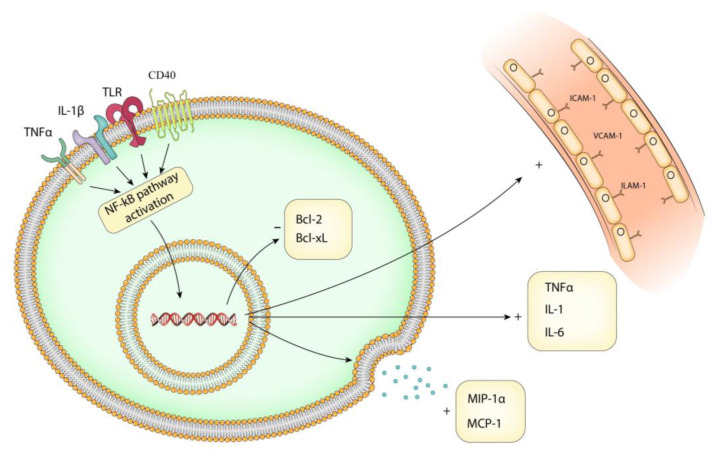
Transcriptional targets of NF-kB pathway. Activated canonical (p65) and non-canonical (p52) NF-kB signaling pathways increase the release of cytokines IL-1β, IL-6, and TNF-α. Adhesion molecules like ICAM-1, VCAM-1, and ECAM-1 have an increased expression in endothelial cells and chemokines (MCP-1, MIP-1α) are upregulated. Anti-apoptotic proteins Bcl-2 and Bcl-xL are downregulated by TNF-α. The illustration is created by the authors using Adobe Illustrator v2020.

## Data Availability

No new data were created or analyzed in this study. Data sharing is not applicable to this article.

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
