# Peer review of "The Role of the NF-kB Pathway in Intracranial Aneurysms"

_brainsci, 2023, doi:10.3390/brainsci13121660_

Round 1

Reviewer 1 Report

Comments and Suggestions for Authors

The paper is a narrative review of the role of NF-kB pathway in the development and rupture of intracranial aneurysms. Moreover, the authors summarize the current knowledge about possible strategies of NF-kB pathway inhibition that may be future research lines for developing therapeutic strategies for intracranial aneurysms.

The following comments should be addressed:

-       The methodology for the review is not explained: the authors should comply with the PRISMA guidelines and find the appropriate checklist to follow.

-       The authors, before exposing the results as they do and summarize the current knowledge, which has an educational value, should precise how many articles they have included and excluded and why so.

-       When mentioning the supposed pathogenesis of development of intracranial aneurysms, see Introduction section, lines 28, 32 and 35, the authors should cite the appropriate updated literature and include the following papers:

1.     Hemodynamic stress, inflammation and intracranial aneurysms development and rupture: a systematic review. World Neurosurg. 2018 Apr 27. pii: S1878-8750(18)30863-5. doi: 10.1016/j.wneu.2018.04.143

2.     Hemodynamic characteristics at the rupture site of cerebral aneurysms: a case study. Neurosurgery. 2012;71:E1202-E1208 [discussion:1209]. 

3.     Biology of intracranial aneurysms: role of inflammation. J Cereb Blood Flow Metab. 2012;32: 1659-1676. 

-        In the Discussion session the authors should consider the current evidences (see the following paper, that should be included in the reference list: Biomechanical Characterization of Intracranial Aneurysm Wall: A Multiscale Study. World Neurosurg. 2018 Nov;119:e882-e889. doi: 10.1016/j.wneu.2018.07.290), indicating that inflammation is the initiating event of intracranial aneurysm development and rupture and its effect is the fragmentation, rarefaction, and loss of network organization of external elastic lamina collagen fibers, observed by microscopy in intracranial aneurysms samples. These ultrastructural changes lead to an upheaval of biomechanical properties of the arterial wall and, under hemodynamic stress forces, aneurysms develop and rupture.

Author Response

Thank you for your feedback related to the writing of the manuscript. Regarding the aspects identified we will answer in detail in the following:

  • The methodology for the review is not explained: the authors should comply with the PRISMA guidelines and find the appropriate checklist to follow.

R: We have restructured the manuscript according to the guidelines and explained out methodology.

  • The authors, before exposing the results as they do and summarize the current knowledge, which has an educational value, should precise how many articles they have included and excluded and why so.

R: We have added a “Results” heading where we explained the process of selection.

  • When mentioning the supposed pathogenesis of development of intracranial aneurysms, see Introduction section, lines 28, 32 and 35, the authors should cite the appropriate updated literature and include the following papers:Hemodynamic stress, inflammation and intracranial aneurysms development and rupture: a systematic review. World Neurosurg. 2018 Apr 27. pii: S1878-8750(18)30863-5. doi: 10.1016/j.wneu.2018.04.143 ; Hemodynamic characteristics at the rupture site of cerebral aneurysms: a case study. Neurosurgery. 2012;71:E1202-E1208 [discussion:1209]. ; Biology of intracranial aneurysms: role of inflammation. J Cereb Blood Flow Metab. 2012;32: 1659-1676. 

R: Introduction has been updated and the references have been included.

  • In the Discussion session the authors should consider the current evidences (see the following paper, that should be included in the reference list: Biomechanical Characterization of Intracranial Aneurysm Wall: A Multiscale Study. World Neurosurg. 2018 Nov;119:e882-e889. doi: 10.1016/j.wneu.2018.07.290), indicating that inflammation is the initiating event of intracranial aneurysm development and rupture and its effect is the fragmentation, rarefaction, and loss of network organization of external elastic lamina collagen fibers, observed by microscopy in intracranial aneurysms samples. These ultrastructural changes lead to an upheaval of biomechanical properties of the arterial wall and, under hemodynamic stress forces, aneurysms develop and rupture.

R: Reference and the subsequent implications have been added to the manuscript.

Reviewer 2 Report

Comments and Suggestions for Authors

The paper requires substantial revisions. As a reviewer, I have several questions and comments regarding your paper. I believe addressing these concerns will help improve the clarity and completeness of your research:

1.      Could you provide more information about the specific methods used to inhibit the NF-kB pathway in the studies mentioned in the review?

2.      How do the different therapeutic strategies for inhibiting NF-kB activation in IA compare in terms of efficacy and safety?

3.      Are there any ongoing clinical trials or potential future developments in this field that you believe hold promise for IA treatment?

4.      Considering the multifaceted nature of IA pathophysiology, do you think a combination of these therapeutic approaches might be more effective than a single intervention?

5.      Can you discuss any potential challenges or limitations associated with targeting the NF-kB pathway for IA treatment, such as off-target effects or long-term safety concerns?

6.      In the context of gene therapy targeting the NF-kB pathway, what specific genes or mechanisms have been explored, and what are the challenges associated with gene delivery to the vascular wall?

7.      Have there been any studies or considerations regarding the timing of intervention in IA development? Is there an optimal stage for implementing these therapeutic strategies?

8.      Could you provide insights into potential patient populations or subtypes of IA that might benefit more from specific therapeutic approaches?

9.      How do these strategies affect other physiological processes or cell functions that are unrelated to IA development, and what are the potential side effects?

10.  What future research directions do you see for IA treatment and the inhibition of the NF-kB pathway, and how might these strategies be integrated into clinical practice?

11.  Are there any biomarkers or imaging techniques that can help identify patients who are more likely to respond to NF-kB pathway-targeted therapies?

12.  Could you elaborate on the role of gender, age, or genetic factors in IA pathophysiology and the potential impact on the effectiveness of these therapeutic strategies?

13.  Is there evidence to suggest that the NF-kB pathway inhibition might be beneficial for preventing IA recurrence in patients who have undergone endovascular coiling or surgical clipping?

14.  How does the interaction between different cell types, such as endothelial cells, vascular smooth muscle cells, and immune cells, influence the NF-kB pathway's role in IA development?

15.  Can you provide insights into the potential challenges associated with translating preclinical findings into clinical treatments for IA?

16.  What is the current state of knowledge regarding the long-term outcomes of patients who have received NF-kB pathway-targeted therapy for IA?

17.  Provide more up-to-date references, especially recent advancements in medicine? Add suitable references to the site with the following PMIDs [PMID: PMID: 37238175, PMID: 37701174].

18.  How do the findings and approaches discussed in this review apply to the treatment of other vascular conditions or neurovascular diseases?

19.  Are there any known interactions or contraindications between NF-kB pathway-targeted therapies and other medications commonly used in IA treatment?

20.  In summary, what key takeaways do you hope researchers and clinicians gain from your review regarding the role of the NF-kB pathway in IA pathophysiology and its potential as a therapeutic target?

Author Response

Thank you for your feedback related to the writing of the manuscript. Regarding the aspects identified in the manuscript we will answer you in detail in the following:

  1. Could you provide more information about the specific methods used to inhibit the NF-kB pathway in the studies mentioned in the review?

Table 2 has been updated and for each compound used to inhibit the NF-kB pathway we added the mechanism of action in the manuscript.

  1. How do the different therapeutic strategies for inhibiting NF-kB activation in IA compare in terms of efficacy and safety?

Regarding efficacy and safety, aspirin is the only drug with preliminary results in patients presenting with IAs and there have been reported no safety issues. We have added details in the text.

  1. Are there any ongoing clinical trials or potential future developments in this field that you believe hold promise for IA treatment?

We have added heading “clinical trials targeting inflammation for treating IA” in response to your question.

  1. Considering the multifaceted nature of IA pathophysiology, do you think a combination of these therapeutic approaches might be more effective than a single intervention?

To our knowledge there are no studies where a combination of therapeutic strategies has been employed, although Aoki et al. has proven that synergic activation of NF-kB pathway with prostaglandin E2 and TNF-α has increased the expression of inflammatory markers. Paragraph was added that explains our position.

  1. Can you discuss any potential challenges or limitations associated with targeting the NF-kB pathway for IA treatment, such as off-target effects or long-term safety concerns?

Safety concern for Etanercept and long-term use of Zinc have been added to the manuscript.

  1. In the context of gene therapy targeting the NF-kB pathway, what specific genes or mechanisms have been explored, and what are the challenges associated with gene delivery to the vascular wall?

The genes explored are Ccl2, Ptgs2, Bcl2 and Bcl-xL. We have added the corresponding genes and what they encode respectively.

  1. Have there been any studies or considerations regarding the timing of intervention in IA development? Is there an optimal stage for implementing these therapeutic strategies?

Regarding timing of intervention most studies choose to start treatment immediately after the aneurysm induction, the day after or one month after. No explanation was given regarding the optimal stage for implementing the therapeutic strategies.

  1. Could you provide insights into potential patient populations or subtypes of IA that might benefit more from specific therapeutic approaches?

Most authors employed similar models for the induction of IA that was first described by Hashimoto et al. in rats. The basic concept involves inducing hemodynamic stress to arterial walls by unilateral common carotid ligation, unilateral renal artery ligation and high-salt diet. The result consists of IA induction at the anterior cerebral artery/olfactory artery bifurcation. To this end, we can conclude that the results are specific for anterior circulation aneurysms however no differences were made between potential patient populations or subtypes of IA in the available research. Paragraph added in the manuscript.

  1. How do these strategies affect other physiological processes or cell functions that are unrelated to IA development, and what are the potential side effects?

The manuscript references studies that employed various cells cultures ranging from aortic endothelial cells, bone marrow mesenchymal stem cells to glioma cells and fibroblasts. Unfortunately, the authors don’t report any side effects of the drugs used in animal testing.

  1. What future research directions do you see for IA treatment and the inhibition of the NF-kB pathway, and how might these strategies be integrated into clinical practice?

Answered in the subheading entitled “Future research directions for treating IA”.

  1. Are there any biomarkers or imaging techniques that can help identify patients who are more likely to respond to NF-kB pathway-targeted therapies?

To the authors knowledge, there are no biomarkers or imaging techniques that are used in the clinical setting that can help identify patients more likely to respond to NF-kB pathway-targeted therapies. Paragraph added in the manuscript.

  1. Could you elaborate on the role of gender, age, or genetic factors in IA pathophysiology and the potential impact on the effectiveness of these therapeutic strategies?

The prevalence of IA is estimated to be 2–5 % in the general population with an equal distribution among sexes and an annual risk of rupture of 1.4%. Ruptured IA have a 30-50% morbidity and a 50% mortality rate posing serious strain on the healthcare system. The therapeutic strategies detailed in this review are currently in an experimental state with only aspirin and statins proceeding to human trials. No differences have been made regarding the impact of gender, age or genetic factors on the effectiveness of treatment. Paragraph added in the manuscript.

  1. Is there evidence to suggest that the NF-kB pathway inhibition might be beneficial for preventing IA recurrence in patients who have undergone endovascular coiling or surgical clipping?

To our knowledge there are no studies reporting whether or not inhibition of NF-kB pathway has been performed after endovascular or surgical treatment. We have added your suggestion to the “future research directions” subheading.

  1. How does the interaction between different cell types, such as endothelial cells, vascular smooth muscle cells, and immune cells, influence the NF-kB pathway's role in IA development?

TNF-α activated endothelial cells stimulate VSMCs and immune cells proliferation and promote early and late apoptosis of VSMCs. This leads to an increase in inflammation in the vascular wall and promotes aneurysm formation. Paragraph added in the manuscript

  1. Can you provide insights into the potential challenges associated with translating preclinical findings into clinical treatments for IA?

Current therapeutic strategies employed have been tested in a preclinical setting, with the exception of statins and aspirin. Potential challenges could rise from translating them to clinical treatments mainly because there is little variability in the preclinical model used, sample sizes are small and the research is conducted under standard conditions, unlike in a clinical scenario. Paragraph added in the manuscript and we have  added a new subheading titled „Limitations"

  1. What is the current state of knowledge regarding the long-term outcomes of patients who have received NF-kB pathway-targeted therapy for IA?

There are limited clinical trials researching the effect of NF-kB pathway inhibition, therefore, to our knowledge, there are no long-term outcomes reported in patients suffering from IA. Paragraph added in the subheading „Limitations”

  1. Provide more up-to-date references, especially recent advancements in medicine? Add suitable references to the site with the following PMIDs [PMID: PMID: 37238175, PMID: 37701174].

References added.

  1. How do the findings and approaches discussed in this review apply to the treatment of other vascular conditions or neurovascular diseases?

The inhibition of the NF-kB pathway proves advantageous in diverse neurovascular disorders like intracranial stenosis, moyamoya disease and ischemic stroke. Consequently, the therapeutic approaches outlined above hold potential for application in managing these specific conditions. Paragraph added in the manuscript

  1. Are there any known interactions or contraindications between NF-kB pathway-targeted therapies and other medications commonly used in IA treatment?

No know interactions or contraindications between NF-kB pathway-targeted therapies and other medications commonly used in IA treatment have been reported. Paragraph added in the manuscript

  1. In summary, what key takeaways do you hope researchers and clinicians gain from your review regarding the role of the NF-kB pathway in IA pathophysiology and its potential as a therapeutic target

Conclusions have been updated.

Round 2

Reviewer 2 Report

Comments and Suggestions for Authors

The paper can be accepted for publication.